# Predictors for Enjoyment in Playing Musical Instruments with a Focus on Psychological Factors

**DOI:** 10.3390/bs15081077

**Published:** 2025-08-07

**Authors:** Weiyi Zhao, Alexander Park, Qian Zhang, Kyung-Hyun Suh

**Affiliations:** 1Department of Interdisciplinary Arts, Sahmyook University, Seoul 01795, Republic of Korea; 2Department of Art Education, Xianda College of Economics and Humanities, Shanghai International Studies University, Shanghai 200089, China; 3Department of Counseling Psychology, Sahmyook University, Seoul 01795, Republic of Korea

**Keywords:** musical instrument, enjoyment, personality, self-directedness, fun-seeking

## Abstract

This study investigated the psychological factors related to the enjoyment of playing musical instruments among Chinese adults. Additionally, it verified a model that can predict enjoyment using psychological variables, demographic profiles, and variables related to music to provide useful information and knowledge for further studies and interventions. The participants were 416 male and female Chinese adults aged 20–68 years. Predictive models were examined using stepwise regression and decision-tree analyses. The results revealed that extraversion, conscientiousness, agreeableness, and hardiness were positively correlated with the enjoyment of playing musical instruments, whereas the behavioral activation system (BAS) and behavioral inhibition system (BIS) showed negative correlations. Stepwise regression analysis revealed that fun-seeking accounted for the greatest variance in enjoyment in playing musical instruments. Fun-seeking, agreeableness, openness, self-directedness, and conscientiousness accounted for approximately 27.2% of the variance in enjoyment in playing musical instruments among Chinese adults. The decision-tree model included enjoyment of music class in childhood, self-directedness, age, experience playing musical instruments, experience growing up in a family that enjoys music, extraversion, and agreeableness. These findings suggest that psychological variables such as fun-seeking and agreeableness may play a more important role in Chinese adults’ enjoyment of playing musical instruments.

## 1. Introduction

Parents often teach their children to play musical instruments beginning at an early age. The reasons are either that they want them to become musicians or that they want their children to be able to play at least one musical instrument ([25]). Given its many advantages, parents may want their children to learn to play musical instruments. Previous studies have emphasized that learning to play a musical instrument may positively impact students’ academic achievement and quality of school life ([5]). However, some scholars argue that these effects may be mediated by pre-existing personal and environmental factors, such as parental involvement, personality traits such as conscientiousness, or access to high-quality instruction, rather than music education per se ([23]).

[24] ([24]) found that students who were learning to play musical instruments showed better cognitive abilities and academic achievement and were more conscientious, open-minded, and ambitious than those who did not. Learning to play a musical instrument was found to be more than twice as effective as learning sports, theater, or dance in enhancing cognitive abilities. Additionally, learning to play a musical instrument may positively affect students’ self-concept ([44]).

Enjoyment playing musical instruments offers benefits even in old age. Numerous studies have shown that active music making enhances older adults’ cognitive, emotional, and social wellbeing ([15], [16]). For example, regular participation in musical activities has been associated with higher levels of subjective wellbeing, self-esteem, and life satisfaction. Moreover, a qualitative meta-synthesis suggested that music education can provide meaningful and empowering experiences later in life ([34]). In addition, playing musical instruments may help prevent cognitive decline, including dementia, by improving verbal memory and the efficiency of the nervous system ([22]; [3]).

[45] ([45]) suggested that learning to play musical instruments not only improves cognitive ability but can also effectively reduce stress by alleviating pressure and creating a sense of stability in daily life. He emphasized the importance of prescribing musical instrument playing as a mental health intervention, as it helps alleviate negative emotions such as anxiety and depression. Music therapy using musical instruments has been used clinically because of its proven efficacy ([41]). According to several studies, musical activities, including playing musical instruments, may positively affect health and promote wellbeing (e.g., [17]). In summary, enjoying playing musical instruments not only provides therapeutic benefits but also enhances overall well-being and life satisfaction.

The present research explored the psychological factors that predict people’s enjoyment of playing musical instruments. We first hypothesized that an individual’s personality is related to the tendency to enjoy playing musical instruments. Personality is a fundamental psychological construct that predicts human behavior and values. We selected the Big-5 personality traits, the most stable personality factors, as predictors ([43]). The Big-5 personality traits are neuroticism, extraversion, conscientiousness, openness, and agreeableness ([14]). In a previous study ([48]), extraversion, openness, and agreeableness were positively correlated with music preference. However, emotional stability, in contrast to neuroticism, was negatively correlated with pursuing the psychological effects of music. [52] ([52]) investigated the relationship between preference for playing a musical instrument and the Big-5 personality traits. Additionally, they found that people who preferred to play melodic instruments, such as the guitar or piano, had higher agreeableness and openness than those who preferred to play rhythm instruments, such as the drums. They also found differences in extroversion depending on the instrument or music genre preferred by individuals. Accordingly, the current study sought to examine whether the Big-5 personality traits predict the tendency to enjoy playing musical instruments.

We also aimed to examine whether temperament or dispositional traits predict the tendency to enjoy playing musical instruments. The behavioral activation system (BAS) and behavioral inhibition system (BIS) were selected as temperamental personalities ([19]). The BAS is related to the dopamine pathway in the brain, sensitive to reward-related cues and actively pursuing it. It is a motivational system that induces emotions such as hope, excitement, happiness, and joy, fostering the expectation that people can achieve their goals ([51]). The BIS is related to the brain’s septum, hippocampus, and serotonin pathway, which is sensitive to cues of punishment and danger, leading to a cessation of behavior ([21]), and is correlated with negative emotions such as anxiety ([54]).

Individuals with high BAS scores are more sensitive to the rewards provided by music and participate more actively in musical activities ([38]). While the BAS has been associated with increased engagement in music-related activities, some researchers distinguish between the reward-reactivity and impulsivity components of the BAS, suggesting that high BAS scores may also reflect trait impulsivity, which can negatively affect sustained learning efforts, such as mastering an instrument ([46]). In a previous study ([48]), all sub-variables of the BAS, including the BIS, were positively correlated with music preference; the Big-5 personality factors accounted for approximately 7% of the variance in music preference, while BAS and BIS accounted for 15%.

Self-efficacy was selected as the psychological variable for predicting the tendency to enjoy playing musical instruments. Self-efficacy is a concept introduced by [7] ([7]) that refers to an individual’s belief in their ability to complete a specific task well and achieve a goal. Self-efficacy is based on the perceived behavioral control included in the theory of planned behavior, which has been shown to predict a wide range of behaviors ([1]). Self-efficacy can predict various behaviors related to goal achievement ([37]). According to [26] ([26]), self-efficacy significantly influences academic achievement. As self-efficacy is important for music education and performance ([57]), there is another concept referred to as music self-efficacy ([55]). However, this study examined whether general self-efficacy, rather than musical self-efficacy, could predict enjoyment when playing musical instruments.

Hardiness may affect a person’s ability to learn and play musical instruments effectively. Introduced by [29] ([29]), psychological hardiness is an inner psychological trait that allows individuals to cope with difficult tasks. People with high commitment, a hardiness factor, find meaning in work or interpersonal relationships and are immersed in their lives ([31]). We assumed that commitment was related to enjoying playing musical instruments. [36] ([36]) emphasized the importance of the commitment of people engaged in music. When performers concentrate on their performance, they can immerse themselves in it, experience flow, play beautifully, and not feel anxious ([40]). Self-directedness or control, which is a hardiness factor, may play an important role in learning musical instruments ([35]). Tenacity, or challenge, is a hardiness trait that approaches adversity experienced in life not as a threat but as an opportunity and approaches it in a challenging manner rather than defensively ([32]; [49]). This trait may be associated with greater enjoyment in playing musical instruments. Accordingly, we assumed that hardy people are more likely to learn musical instruments, successfully acquire performance skills, and enjoy playing instruments.

This study aimed to investigate how personality, temperament, self-efficacy, and hardiness are related to individuals’ enjoyment of playing musical instruments and to verify models that can predict the enjoyment of playing musical instruments. To achieve this purpose, the following research questions were examined: First, are there significant relationships between the Big-5 personality factors, BAS/BIS, self-efficacy, and hardiness and the enjoyment of playing musical instruments among Chinese adults? Second, what is the appropriate stepwise regression model for predicting Chinese adults’ enjoyment of playing musical instruments? Third, what is the appropriate decision-tree model for predicting Chinese adults’ enjoyment of playing musical instruments?

Understanding who is likely to enjoy playing musical instruments can help identify suitable candidates for instrumental training and enhance the effectiveness of such programs. Previous research has explored the predictors of participation in music lessons based on parental traits ([12]). Anticipating musical enjoyment may also inform efforts to enhance wellbeing through music. Such insight can help parents and educators avoid pressuring individuals who are unlikely to enjoy learning an instrument and guide the appropriate use of music-based interventions, such as therapy.

## 2. Materials and Methods

### 2.1. Participants

A total of 416 Chinese adults participated in the study. The participants’ ages ranged from 20 to 68 years, with a mean age of 38.59 ± 13.15 years. Detailed demographic characteristics of the participants are provided in Section 3.

### 2.2. Participants’ Characteristics

Of the participants, 203 (48.8%) were men and 213 (51.2%) were women (Table 1). In addition, 148 were in their 20s (35.6%), 58 were in their 30s (13.9%), 113 were in their 40s (27.2%), 58 were in their 50s (13.9%), and 39 were in their 60s (9.4%). A total of 62 (14.9%) participants reported living alone.

Of the participants, 71 (17.1%) had majored or were majoring in music, and 190 (45.7%) reported proficiency in playing at least one instrument. Additionally, 87 (20.9%) participants reported growing up in a family that enjoyed music, 190 (45.7%) enjoyed music classes during childhood or adolescence, and 190 (45.7%) reported having experience playing musical instruments with their families.

### 2.3. Data Collection

Data were collected from the Wenjuanxing website (wjx.cn), an online survey platform in China. Ethical approval was obtained from the institutional review board (IRB), and written informed consent was acquired from all participants online. Every effort was made to ensure that the data collection process adhered to ethical standards.

Participants were informed that they could withdraw from the survey at any time if they experienced discomfort while responding. They were also assured that all data would be used exclusively for research purposes and securely stored on an encrypted computer for three years before being permanently deleted.

To recruit participants, we promoted the study through online bulletin boards and social networking sites (SNSs). Additionally, to ensure the inclusion of individuals with musical experience, we actively invited adults proficient in playing musical instruments.

### 2.4. Instruments

#### 2.4.1. Enjoyment in Playing Musical Instrument

The participants’ enjoyment of playing musical instruments was measured using the Enjoyment of Playing Instruments Scale developed by [59] ([59]). This scale measures the following three aspects of enjoyment derived from playing an instrument: learning and social bonds, achievement and pride, and cognitive refreshment and stimulation. It consists of 16 items, and each item was rated on a 5-point Likert scale (1: strongly disagree to 5: strongly agree). A scale-development study ([59]) showed a stable factorial structure, relatively good test–retest reliability, internal consistency, and criterion-related validity. In this study, the Cronbach’s α coefficients were 0.92 for learning and social bonds, 0.93 for achievement and pride, 0.87 for cognitive refreshment and stimulation, and 0.97 for all items.

#### 2.4.2. Personality

Participants’ personalities were measured using the Chinese Big Five Personality Inventory-15 (CBF-PI-15: [60]). This scale measures five personality factors based on the Big-5 personality concept: neuroticism, extraversion, conscientiousness, openness, and agreeableness. The CBF-PI-15 consists of 15 items, with 2 items reverse-scored. Each item is rated on a 6-point Likert scale (1: strongly disagree to 6: strongly agree). A scale-development study ([60]) demonstrated good reliability and validity. Although some factors showed relatively low internal consistency, this was acceptable considering the nature of Cronbach’s α, which is sensitive to the number of items. In this study, the Cronbach’s αs were 0.71 for neuroticism, 0.66 for extraversion, 0.66 for conscientiousness, 0.55 for openness, and 0.68 for agreeableness.

#### 2.4.3. BAS/BIS

The BAS and BIS were measured using [9]’s ([9]) BAS/BIS Scale. We used the Chinese version of the scale validated by [10] ([10]). This scale consists of 18 items categorized into three subscales, including reward responsiveness (four items), drive (four items), and fun-seeking (five items) that reflect behavioral activation or incentive responsiveness and one sub-scale (five items) that reflects behavioral inhibition sensitivity or threat responsiveness. Each item is rated from 1 (strongly agree) to 4 (strongly disagree). All items were reverse-scored so that higher scores indicated higher BAS and BIS levels. The Cronbach’s αs in this study were 0.77–0.81 for the subscales of the BAS and 0.79 for the BIS.

#### 2.4.4. Self-Efficacy

Self-efficacy was measured using the Chinese version of the General Self-Efficacy Scale developed by [58] ([58]). This scale consists of 10 items that assess self-efficacy, reflecting individuals’ belief in their ability to handle and overcome life’s challenges. Examples of items include “I can always manage to solve difficult problems if I try hard enough” and “I can usually handle whatever comes my way.” Each item is rated on a 4-point Likert scale ranging from 1 (not at all true) to 4 (exactly true). The Cronbach’s α of the items was 0.77 in this study.

#### 2.4.5. Hardiness

Psychological hardiness was measured using the Brief Measure of Hardiness developed by [49] ([49]). This scale consists of 12 items and three subscales: commitment (4 items), self-directedness (4 items), and tenacity (4 items). Each item is rated on a 6-point Likert scale ranging from 1 (not at all true) to 6 (very true), with higher scores indicating a hardier person. Examples of items include “When I open my eyes in the morning, I look forward to the day,” “My decisions shape my life,” and “Adversity makes me grow.” In this study, the Cronbach’s αs of the items was 0.76 for commitment, 0.72 for self-directedness, 0.70 for tenacity, and 0.89 for all items.

### 2.5. Statistical Analysis

All data were analyzed using the Statistical Package for the Social Sciences (SPSS) for Windows version 26. Prior to performing parametric statistical analysis, the skewness and kurtosis of the psychological variables were examined to assess normality. Correlation and stepwise regression analyses were conducted using parametric statistics, and decision-tree analyses were conducted using nonparametric statistics.

The chi-square automatic interaction detection (CHAID) technique was used for decision-tree analysis. [27] ([27]) developed this technique, an algorithm that performs multiple separations based on the chi-square (*χ*^2^) from cross-tabulations and the F-statistic from analysis of variance. The total expected score was selected as the target variable, and because it is a continuous variable, the likelihood ratio *χ*^2^ statistic was used. The maximum number of levels was set to three, and the minimum number of cases for the parent and child nodes was set to 30 and 10, respectively.

## 3. Results

### 3.1. Relationships Between Personality, BAS/BIS, Self-Efficacy, and Hardiness and the Enjoyment of Playing Instruments

Table 2 shows the results of the correlation analysis of the Big-5 personality traits, BAS/BIS, self-efficacy, and hardiness with enjoyment in playing instruments among Chinese adults. Because the absolute values of skewness and kurtosis of all variables did not exceed 2.0, the data did not significantly deviate from normality, indicating that parametric statistical analysis was appropriate ([28]).

Correlation analysis showed that extraversion (*r* = 0.16, *p* < 0.001), conscientiousness (*r* = 0.14, *p* < 0.01), and agreeableness (*r* = 0.24, *p* < 0.001) were positively correlated with enjoyment in playing musical instruments, whereas neuroticism and openness were not significantly correlated. In contrast, both BAS (*r* = −0.34, *p* < 0.001) and BIS (*r* = −0.31, *p* < 0.001) were negatively correlated with enjoyment in playing musical instruments. All sub-factors of the BAS, namely reward responsiveness (*r* = −0.29, *p* < 0.001), drive (*r* = −0.31, *p* < 0.001), and fun-seeking (*r* = −0.33, *p* < 0.001), were significantly correlated with the enjoyment of playing musical instruments.

Additionally, hardiness was positively correlated with the enjoyment in playing instruments (*r* = 0.23, *p* < 0.001), whereas self-efficacy was not significantly correlated (*r* = 0.06, n.s.). All sub-factors of hardiness, commitment (*r* = 0.17, *p* < 0.001), self-directedness (*r* = 0.25, *p* < 0.001), and tenacity (*r* = 0.22, *p* < 0.001) were positively correlated with the enjoyment of playing musical instruments.

### 3.2. Predictive Models for Enjoyment in Playing Musical Instruments

We examined models that predicted the playing of musical instruments among Chinese adults. First, we conducted a stepwise regression analysis with psychological variables or subfactors, including a correlation analysis. Multicollinearity problems occur when tolerance is less than 0.2 and the variance inflation factor (VIF) is greater than 5.0 ([53]). Because tolerance of the predictors included in the stepwise regression model was 0.444–0.586 and the VIFs were 1.707–2.253, the multicollinearity was not significant.

As shown in Table 3, fun-seeking was the strongest predictor of the enjoyment of playing musical instruments (*β* = −0.330, *p* < 0.001), followed by agreeableness (*β* = 0.280, *p* < 0.001), openness (*β* = 0.208, *p* < 0.001), self-directedness (*β* = 0.250, *p* < 0.001), drive (*β* = −0.186, *p* < 0.01), and conscientiousness (*β* = 0.143, *p* < 0.05) in this stepwise regression model. Fun-seeking alone accounted for approximately 10.9% of the variance in the enjoyment of playing musical instruments.

In addition to fun-seeking, agreeableness accounted for approximately 7.7% of the variance in the enjoyment of playing instruments and openness accounted for approximately 3.0%. Self-directedness accounted for an additional 3.4% of the variance in enjoyment in playing instruments, except for the effects of fun-seeking, agreeableness, and openness. Notably, openness was not significantly correlated with the enjoyment of playing musical instruments in the bivariate correlation analysis, but it emerged as a significant predictor in the stepwise regression model when controlling for other variables.

To examine the decision-tree model that predicts the enjoyment of playing musical instruments among Chinese adults, the variables and sub-factors, including those used in the correlation analysis, demographic characteristics, parametric and non-parametric factors, as well as variables related to experiences and environment surrounding instrument playing, were entered as potential predictors.

The results revealed a total of 18 nodes, including 11 terminal nodes, with a tree depth of three (Figure 1). The risk estimate was 109.30 (*SE* = 10.39), and the average risk estimate of the 10-fold cross-validation was 147.16 (*SE* = 14.34), indicating differences within the margin of error.

The average root node enjoyment of playing instruments was 48.83. Nine nodes (Nodes 1, 4, 5, 6, 9, 10, 11, 12, and 15) exceeded this average, and Chinese adults belonging to these nodes showed higher scores on the enjoyment of playing instruments (Figure 1). The order of gain nodes was 12 (9.4%), 10 (2.6%), 11 (15.1%), 9 (11.3%), 6 (7.9%), 15 (2.9%), 3 (6.3%), 13 (25.2%), 16 (6.3%), 14 (9.9%), and 17 (3.1%; Table 4).

The first criterion used to classify the level of enjoyment in playing musical instruments was whether the participants enjoyed music classes during childhood (Figure 1). Among the 186 participants who responded affirmatively, the average enjoyment score was relatively high at 60.77 (Node 1). Within this group, those with self-directedness scores above 18 reported even higher enjoyment, averaging 65.95 (Node 5). Of these, individuals who also grew up in families that appreciated music showed the highest enjoyment scores, reaching 69.95 (Node 11).

Participants with favorable early music experiences and self-directedness scores between 14 and 18 still showed relatively high enjoyment levels (*M* = 58.66; Node 4). In this subgroup, those who had played instruments with family members scored higher, averaging 64.55 (Node 10).

In contrast, among the 230 participants who did not report enjoying music classes during childhood, the overall enjoyment score was lower, at 39.17 (Node 2). However, within this group, younger participants (aged 31 years or below) demonstrated a comparatively higher enjoyment score of 55.97 (Node 6).

Among those aged 31–58 years with no early positive music experience, the average score dropped to 37.27 (Node 7). Nevertheless, individuals in this subgroup with agreeableness scores above 16 reported moderately higher enjoyment, at 53.50 (Node 15).

Finally, among participants over 58 years of age who had not enjoyed early music classes, the average enjoyment was low at 32.69 (Node 8). Notably, those with extraversion scores above 15 in this group showed the lowest overall enjoyment level, averaging just 26.08 (Node 17).

When the commitment score was between 11 and 15 and the empathy score was over 13, the average of expectations for the future was 27.10 (Node 13). Married participants had higher average expectations for the future (30.07; node 19). However, when the commitment score was between 11 and 15 and the empathy score was over 13, the average of expectations for the future was at 22.20; young adults with a self-directedness score of over 16 had a higher average for expectations for the future at 24.05 (Node 18).

Even if the commitment score was between 9 and 11, if young adults perceived themselves as healthy, the average of their expectations for their future was 22.50 (Node 10). If the commitment score was 9 or lower and the stress score was over 37, the average expectation for the future was lowest at 10.76 (Node 9).

## 4. Discussion

The current study explored variables that could predict the enjoyment of playing musical instruments among Chinese adults to provide useful information and knowledge for future studies on music education or therapy. Psychological factors associated with the enjoyment of playing musical instruments, such as personality, BAS/BIS, self-efficacy, and hardiness, were investigated using a stepwise regression model. Additionally, demographic profiles, including categorical data potentially linked to enjoyment in playing musical instruments, were incorporated as predictors in the decision-tree model. The implications of these findings are as follows.

The more agreeable Chinese adults were, the more likely they were to enjoy playing musical instruments. Among the Big-5 personality traits, agreeableness had the highest accountability for enjoyment of playing musical instruments at 5.7% (*r* = 0.239). Agreeableness was the determinant predictor in the stepwise regression model predicting Chinese adults’ enjoyment of playing musical instruments and was also included in the decision-tree model. In a previous study ([6]), agreeableness was significantly correlated with trait empathy for rhythmic entrainment during spontaneous movement in response to music. These results suggest that empathizing with the emotions that performers feel when playing an instrument may be the key to enjoying it.

The more extraverted Chinese adults were, the more likely they were to enjoy playing musical instruments. [52] ([52]) found that extraverts prefer to listen to instrumental music, whereas our study found that they tend to enjoy playing musical instruments. Both amateur and professional instrumentalists show lower levels of extraversion than singers or vocalists ([33]). The current results thus have educational and clinical implications. For example, teaching extraverted children or adolescents how to professionally play musical instruments may be effective. It may also be effective if an extraverted adult learns to play a musical instrument as a hobby. Additionally, the application of music therapy to extraverts may be effective; however, this requires verification in future studies.

The higher the conscientiousness of Chinese adults, the more likely they are to enjoy playing musical instruments. Conscientiousness plays an important role in academic achievement and training performance ([18]). [13] ([13]) believed that conscientiousness affects children’s and adults’ adaptability and achievement in musical instrument training and lessons. As conscientiousness is the personality trait of being responsible, diligent, dutiful, self-disciplined, and dependable ([50]), conscientious people tend to be more efficient in most learning situations. Therefore, it may be effective for teachers to provide instrumental lessons to individuals with high levels of conscientiousness.

Openness was found to have no significant relationship with enjoyment of playing musical instruments in the correlation analysis but was included in the stepwise regression model. In this model, openness accounted for approximately 3% of the variance in the enjoyment of playing musical instruments, in addition to fun-seeking and agreeableness. This result suggests that when the effects of fun-seeking and agreeableness are adjusted, openness can positively influence the enjoyment of playing musical instruments. In a study by [33] ([33]), musicians showed higher openness than non-musicians. The current study suggests that people with high openness are more likely to enjoy playing musical instruments when the effects of fun-seeking and agreeableness are controlled. However, openness itself was not significantly correlated with enjoyment in playing musical instruments, and the relationship between these two variables requires further in-depth study. For example, further studies could examine whether openness interacts with other traits such as agreeableness or conscientiousness to affect music-related behaviors.

The results showed that Chinese adults with higher BAS and BIS scores tended to enjoy playing musical instruments more. [20] ([20]) described the BAS and BIS as playing a role in balancing whether to perform or quit a behavior. However, both the BAS and BIS showed negative relationships with the enjoyment of playing musical instruments. Because the BAS and BIS can both have positive and negative correlations with certain psychological variables ([11]; [42]), this result cannot be considered contradictory. A stepwise regression model revealed that higher levels of fun-seeking were associated with lower enjoyment in playing musical instruments among Chinese adults. Because individuals with higher levels of fun-seeking tend to be more attracted to stimulating activities ([9]), this result may suggest that such individuals do not enjoy playing musical instruments, as it requires sustained practice and does not provide immediate gratification. Alternatively, individuals with high BAS or BIS scores may prefer activities that provide immediate stimulation or avoid frustration, which structured instrumental training may not immediately offer. Further studies should explore whether musical improvisation or performance-based tasks elicit different levels of enjoyment in these individuals.

Additionally, drive, a factor in the BAS, was included in the stepwise regression model, suggesting that people may not enjoy playing musical instruments if they are attracted to other fun activities and have a strong drive to engage in them. Fun-seeking was the most decisive predictor in the stepwise regression model of Chinese adults’ enjoyment in playing musical instruments. In particular, the effect size of fun-seeking on enjoyment in playing musical instruments was relatively high, at 0.109. Our analysis suggests that individuals’ tendencies to seek fun should be considered in educational and therapeutic contexts involving musical instrument instruction. For example, individuals with high fun-seeking tendencies might be less inclined to sustain long-term engagement in musical instrument training. Therefore, educators and therapists might consider integrating more engaging or stimulating elements into early-stage instruction. However, further empirical studies are required to determine whether such individuals benefit less from traditional approaches to music education and therapy.

Contrary to this hypothesis, general self-efficacy was not significantly correlated with enjoyment in playing musical instruments among Chinese adults. In other words, general self-efficacy did not affect the enjoyment of playing musical instruments. This finding suggests that general self-efficacy does not significantly predict enjoyment in playing instruments, and this may indicate that music-specific self-efficacy is more relevant. Further studies should differentiate between general and domain-specific self-efficacy when examining musical engagement. However, these implications are limited to general self-efficacy, and the efficacy of playing musical instruments may differ. Thus, it is necessary to explore the relationship between musical performance efficacy and enjoyment in playing musical instruments in the future.

In the current study, the hardier the Chinese adults were, the more they enjoyed playing musical instruments. Self-directedness was included in both stepwise regression and decision-tree models to predict Chinese adults’ enjoyment in playing musical instruments. Self-directedness emerged as a predictor in both the stepwise regression and decision-tree models, indicating that the more self-directed Chinese adults were, the more they enjoyed playing musical instruments. The more self-directed Chinese adults were, the more they enjoyed musical instruments. Self-directed learning is important in adult music learning ([35]). Our analysis suggests that self-directed tendencies may be decisive in adults’ enjoyment of playing musical instruments. The findings demonstrate that teaching self-directed adults how to play musical instruments or incorporating instrument-based music therapy may be more effective.

While we assumed that individuals with higher levels of hardiness tend to enjoy playing musical instruments, it is also worth considering the reverse possibility: whether the discipline and perseverance required to learn an instrument might, in turn, foster psychological hardiness. Long-term musical training involves coping with challenges, maintaining self-directed practice, and persisting through frustration. These efforts reflect qualities that overlap with the dimensions of hardiness, such as commitment, control, and challenge ([30]). Future longitudinal research should explore whether structured musical training contributes to the development of hardy traits, particularly among younger learners and clinical populations.

Additionally, in the decision-tree model, whether Chinese adults enjoyed music classes during childhood was the most decisive predictor of enjoyment in playing musical instruments. The importance of music education in childhood has been emphasized, and many musicians have stated that they enjoyed music classes from a young age ([47]), revealing that adults who enjoyed music classes from an early age are more likely to enjoy musical instruments in adulthood. Family also plays an important role in shaping adults’ musical engagement. It is known that the family plays an important role in music learning, including learning to play a musical instrument ([4]; [56]). Our findings suggest that adults who grew up in families that enjoyed music or played instruments together are more likely to enjoy playing musical instruments. Younger participants in this study also reported higher enjoyment levels, although this age-related trend requires further investigation to confirm its consistency.

In this study, we use “music therapy” broadly to refer to the psychological application of instrument playing; however, the term encompasses a range of diverse approaches. Clinical music therapy involves individualized interventions by licensed professionals, targeting goals such as anxiety reduction or communication enhancement ([8]). In contrast, community music therapy emphasizes inclusive participatory music making that fosters social connection and empowerment regardless of diagnosis ([2]). [39] ([39]) proposed a broader framework in which music supports emotional regulation, cognitive engagement, and social wellbeing. These perspectives highlight the multifaceted therapeutic roles of music and suggest the need for greater clarity when applying or evaluating instrument-based interventions.

This study had several limitations in interpreting and concluding its findings. First, the participants in the online survey were not representative of all Chinese adults. However, because the survey was conducted online, the limitations of specific regions were overcome. Second, while we discussed potential cause-and-effect relationships between variables based on previous studies and logical reasoning, such conclusions cannot be definitively drawn from correlational data alone and require experimental validation. Third, although decision-tree analysis has various advantages over other parametric inferential statistical analyses, there are limitations in using SPSS. Despite these limitations, this study may stimulate further research on the enjoyment of playing musical instruments and provide useful information and knowledge for music education and interventions in music therapy using musical instrument playing.

## 5. Conclusions

We found that extraversion, conscientiousness, agreeableness, BAS, BIS, and hardiness were significantly correlated with the enjoyment in playing musical instruments among Chinese adults. Additionally, fun-seeking, agreeableness, openness, and self-directedness were notable predictors of adults’ enjoyment of playing musical instruments. The decision-tree model further identified variables such as enjoyment of music classes in childhood, self-directedness, age, family music experiences, and familial musical atmosphere as being associated with adults’ musical enjoyment. These results indicate patterns of association rather than causation, as this study employed a correlational design.

Although traits such as fun-seeking and the BAS/BIS were negatively associated with musical enjoyment, these findings should be interpreted with caution. Individuals scoring high on these traits may find greater satisfaction with less conventional musical activities, such as improvisation, ensemble play, or high-stimulation musical environments. However, these hypotheses require empirical validation. Importantly, the cross-sectional nature of the data limits causal inferences, and reverse or reciprocal relationships cannot be ruled out.

Despite these limitations, the findings provide insights that may help music educators and therapists tailor their approaches more effectively. For example, considering individuals’ early music exposure, self-directedness, and motivational tendencies may enhance learner engagement and educational outcomes. However, practical applications should be grounded in further longitudinal or experimental research to more clearly disentangle causality from correlation.

## Figures and Tables

**Figure 1 behavsci-15-01077-f001:**
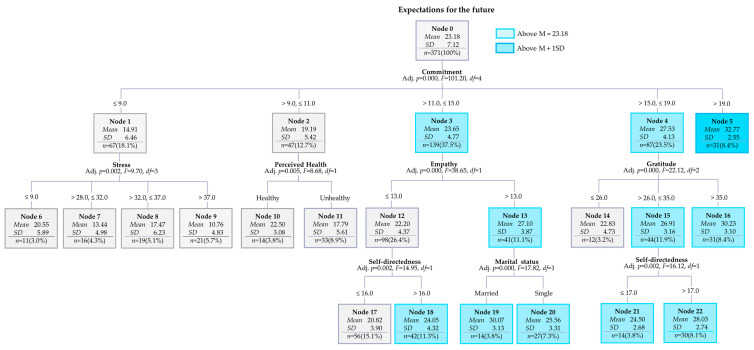
Decision-tree model of expectations for the future among young Korean adults.

**Table 1 behavsci-15-01077-t001:** Characteristics of participants (*N* = 416).

Variables		Frequency	Percent (%)
Gender	Man	203	48.8
	Woman	213	51.2
Age	20s	148	35.6
	30s	58	13.9
	50s	113	27.2
	60s	58	13.9
	70s	39	9.4
Housing type	Living alone	62	14.9
	Living with other(s)	354	85.1
Majoring or majored in music?	Yes	71	17.1
	No	345	82.9
Can you play an instrument well?	Yes	190	45.7
	No	226	54.3
Grew up in a family that enjoys music?	Yes	87	20.9
	No	329	79.1
Enjoyed music classes?	Yes	186	44.7
	No	230	55.3
Played instrument(s) with family?	Yes	83	20
	No	333	80

**Table 2 behavsci-15-01077-t002:** Correlational matrix for Big-5 personality traits, BAS/BIS, self-efficacy, hardiness, and enjoyment of playing instruments (*N* = 416).

Variables	1	2	3	4	5	6	6-1	6-2	6-3	7	8	9	9-1	9-2	9-3	10
1. Neuroticism	1															
2. Extraversion	0.41 ***	1														
3. Conscientiousness	0.42 ***	0.65 ***	1													
4. Openness	−0.55 ***	−0.58 ***	−0.59 ***	1												
5. Agreeableness	0.40 ***	0.56 ***	0.5 4***	−0.51 ***	1											
6. BAS	0.06	0.09	−0.15 **	0.19 ***	0.13 *	1										
6-1. Reward responsiveness	−0.04	0.06	−0.11 *	0.17 ***	0.12 *	0.02	1									
6-2. Drive	0.07	0.09	−0.14 **	0.16 **	0.11 *	0.90 ***	0.75 ***	1								
6-3. Fun-seeking	0.12 *	0.10	−0.15 **	0.18 ***	0.12 *	0.92 ***	0.75 ***	0.74 ***	1							
7. BIS	0.15 **	0.14 **	−0.16 **	0.16 **	0.13 *	0.76 ***	0.68 ***	0.65 ***	0.73 ***	1						
8. Self-efficacy	0.42 ***	0.44 ***	0.44 ***	−0.43 ***	0.45 ***	−0.01	0.03	−0.03	0.01	0.05	1					
9. Hardiness	0.38 ***	0.57 ***	0.63 ***	−0.57 ***	0.67 ***	−0.01	0.04	0.01	0.02	0.12 *	0.62 ***	1				
9-1. Commitment	0.34 ***	0.51 ***	0.56 ***	−0.51 ***	0.58 ***	−0.07	−0.08	−0.06	−0.04	0.06	0.57 ***	0.91 ***	1			
9-2. Self-directedness	0.34 ***	0.50 ***	0.58 ***	−0.51 ***	0.63 ***	0.02	−0.03	0.04	0.04	0.13 **	0.55 ***	0.90 ***	0.72 ***	1		
9-3. Tenacity	0.34 ***	0.53 ***	0.58 ***	−0.52 ***	0.60 ***	0.04	0.01	0.06	0.06	0.13 **	0.58 ***	0.91 ***	0.74 ***	0.74 ***	1	
10. Enjoyment of playing instruments	−0.08	0.1 6 **	0.14 **	0.08	0.24 ***	−0.34 ***	−0.29 ***	−0.31 ***	−0.33 ***	−0.31 ***	0.06	0.23 ***	0.17 ***	0.25 ***	0.22 ***	1
*M*	13.17	13.20	13.52	7.30	13.57	37.11	11.68	11.42	14.01	13.88	28.75	53.94	18.04	18.02	17.88	48.83
SD	3.06	2.84	2.74	2.63	2.84	8.18	2.94	2.82	3.22	3.30	4.60	9.86	3.74	3.61	3.33	17.05
Skewness	−1.09	−0.84	−0.91	0.88	−1.06	0.71	0.82	0.63	−0.61	0.59	−0.45	−1.05	−0.92	−0.88	−0.77	0.29
Kurtosis	1.60	1.29	1.82	1.44	1.82	−0.32	−0.18	−0.35	−0.26	−0.28	0.92	2.85	1.40	1.71	1.78	−1.45

* *p* < 0.05, ** *p* < 0.01, *** *p* < 0.001.

**Table 3 behavsci-15-01077-t003:** Results of the stepwise regression analysis of enjoyment in playing instruments.

Variables	*β*	*t*	∆*R*^2^	*F*
Fun-seeking	−0.330	−7.12 ***	0.109	26.38 ***
Agreeableness	0.280	6.27 ***	0.077	
Openness	0.208	4.09 ***	0.030	
Self-directedness	0.250	4.37 ***	0.034	
Drive	−0.186	−2.94 **	0.013	
Conscientiousness	0.143	2.47 *	0.009	

* *p* < 0.05, ** *p* < 0.01, *** *p* < 0.001.

**Table 4 behavsci-15-01077-t004:** Gain summary for nodes.

Nodes	*N*	%	*M*
12	39	9.4	69.95
10	11	2.6	64.55
11	63	15.1	63.48
9	47	11.3	57.28
6	33	7.9	55.97
15	12	2.9	53.50
3	26	6.3	45.15
13	105	25.2	37.22
16	26	6.3	36.00
14	41	9.9	32.63
17	13	3.1	26.08

Growing method: CHAID.

## Data Availability

The datasets used and analyzed in this study can be obtained from the corresponding author upon reasonable request.

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
