# Peer review of "Predictors for Enjoyment in Playing Musical Instruments with a Focus on Psychological Factors"

_behavsci, 2025, doi:10.3390/bs15081077_

Round 1

Reviewer 1 Report

Comments and Suggestions for Authors

The article clearly presents its objectives and methods, as well as a wide range of relevant prior studies and theoretical frameworks. However, the literature review is occasionally descriptive rather than analytical. Additionally, some references are used to assert value rather than critically contextualize and discuss the main subject of the article. 

The arguments and discussion of findings could be improved. While the discussion logically follows the results and includes thoughtful interpretations, certain claims (e.g., the inefficacy of music education for individuals with high fun-seeking scores) would benefit from a more cautious, evidence-based tone. Engaging more with alternative explanations would improve the balance of the text. Although the article includes a solid range of references, the literature review could benefit from more critical engagement. Sometimes sources are cited to support general claims (such as ideas that "music improves well-being" or  that "BAS influences musical activity") without deeper analysis or discussion of contrasting perspectives.

The literature review would benefit from additional studies, as those by Andrea Creech,  on music activities for older adults.
Just three examples:

Creech, A., et al. (2013). The power of music in the lives of older adults. Research Studies in Music Education, 35(1).

Creech, A., et al.  (2013). Active music making: a route to enhanced subjective well-being among older people. Perspectives in Public Health, 133(1).

Laes, T., & Creech, A. (2023). Re-conceptualizing music education in the older adult life course: A qualitative meta-synthesis. International Journal of Population Studies, 9(3).

This could enrich the theoretical framework and strengthen the argument.

The conclusions generally reflect the findings, but a few statements overreach the correlational nature of the study. The distinction between association and causation should be made clearer, and the limitations deserve more consideration when forming the final claims.

Comments on the Quality of English Language

The English is generally understandable, but there are recurrent issues with  clarity and sometimes with grammar also. 

Some examples:

In the five paragraphs after the table, number 4 (within 3.2), the phrase "(did not) enjoy music classes in childhood",  is repeated eight times, making the text "heavy" in some sense.

Another expression that is repeated too much is "this study." While understandable, it does not help the text's fluidity.

The paragraph just before 2. "Material and Methods", starting with If we can predict individuals’ ...", is wordy and repetitive, making it confusing. 

In section 3.2, the sentence ""Openness may positively affect the enjoyment of playing musical instruments in the regression model, although there was no significant correlation between openness and the enjoyment of playing
instruments in the correlation analysis", is  logically confusing; clarify what you want to say.

Same on the page 13 (within the Discussion): "For example, teaching children or adolescents with high fun seeking to play musical instruments or music therapy using musical instruments for individuals with high fun seeking may be ineffective." This is a strange sentence.

I consider that a professional language revision is needed to improve the clarity and impact of the manuscript.

Author Response

Response to Reviewer 1 Comments

Thank you very much for taking the time to review this manuscript. Please find the detailed responses below and the corresponding revisions/corrections highlighted/in track changes in the re-submitted files. The revised parts were marked in red, and we included the page and line of the revised part. We appreciate your complimentary comments. We have omitted our response to your kind words here.

Point-by-point response to Comments and Suggestions for Authors

Comments 1: The arguments and discussion of findings could be improved. While the discussion logically follows the results and includes thoughtful interpretations, certain claims (e.g., the inefficacy of music education for individuals with high fun-seeking scores) would benefit from a more cautious, evidence-based tone. Engaging more with alternative explanations would improve the balance of the text.

Response 1: Thank you for your valuable comments. We have carefully revised the Discussion section to address your suggestions regarding the strength of some claims and the need for more balanced interpretations. Specifically, we have modified the wording related to fun-seeking tendencies to avoid implying deterministic or overly strong conclusions. Instead, we emphasized that individuals with high fun-seeking scores may prefer more engaging or novel music learning approaches, and that further empirical research is needed to confirm the efficacy of instructional or therapeutic strategies in such populations. [Line 434-439]

Additionally, we expanded the discussion of alternative explanations regarding the roles of BAS and BIS, suggesting that high scores on these scales may reflect a preference for immediate rewards or aversion to structured tasks, rather than a general aversion to music itself. [Line 422-426, 511-517]

We also included interpretive clarification regarding openness and self-efficacy, noting the potential role of interaction effects and the distinction between general and music-specific self-efficacy. These revisions aim to enhance the depth and nuance of our discussion while maintaining scientific rigor and cautious interpretation. [Line 409-411, 442-446]

We greatly appreciate your helpful feedback, which significantly improved the clarity and balance of our manuscript.

For example, individuals with high fun-seeking tendencies might be less inclined to sustain long-term engagement in musical instrument training. Therefore, educators and therapists might consider integrating more engaging or novelty-rich elements into early-stage instruction. However, further empirical studies are required to determine whether such individuals benefit less from traditional approaches to music education and therapy.

Alternatively, individuals with high BAS or BIS scores may prefer activities that provide immediate stimulation or avoid frustration, which structured instrumental training may not immediately offer. Further studies should explore whether musical improvisation or performance-based tasks elicit different levels of enjoyment in these individuals.

Although traits such as fun-seeking and the BAS/BIS were negatively associated with musical enjoyment, these findings should be interpreted with caution. Individuals scoring high on these traits may find greater satisfaction with less conventional musical activities, such as improvisation, ensemble play, or high-stimulation musical environments. However, these hypotheses require empirical validation. Importantly, the cross-sectional nature of the data limits causal inferences, and reverse or reciprocal relationships cannot be ruled out.

For example, further studies could examine whether openness interacts with other traits such as agreeableness or conscientiousness to affect music-related behaviors.

This finding suggests that general self-efficacy does not significantly predict enjoyment in playing instruments, indicating that music-specific self-efficacy may be more relevant. Further studies should differentiate between general and domain-specific self-efficacy when examining musical engagement.

Comments 2: Although the article includes a solid range of references, the literature review could benefit from more critical engagement. Sometimes sources are cited to support general claims (such as ideas that "music improves well-being" or that "BAS influences musical activity") without deeper analysis or discussion of contrasting perspectives.

Response 2: Thank you for your valuable comment on our literature review. Following your suggestion, we have expanded the review to include more critical engagement with existing sources. For example, we contrasted the positive effects of musical training with alternative interpretations related to pre-existing personality traits and educational environments. We also elaborated on conflicting findings regarding the role of BAS in musical engagement, highlighting the importance of contextual moderators: [Line 33-38, 93-97]

Previous studies have emphasized that learning to play a musical instrument may positively impact students’ academic achievement and quality of school life (Baker et al., 2023; Shi, 2022). However, some scholars argue that these effects may be mediated by pre-existing personal and environmental factors, such as parental involvement, personality traits such as conscientiousness, or access to high-quality instruction, rather than music education per se (Hallam, 2010).

While the BAS has been associated with increased engagement in music-related activities, some studies differentiate between the reward-reactivity and impulsivity components of the BAS, suggesting that high BAS scores might also reflect trait impulsivity, which can negatively affect sustained learning efforts, such as mastering an instrument (Smillie et al., 2006).

Hallam, S. (2010). The power of music: Its impact on the intellectual, social and personal development of children and young people. International Journal of Music Education, 28(3), 269–289. https://doi.org/10.1177/0255761410370658

Smillie, L. D., Jackson, C. J., & Dalgleish, L. I. (2006). Conceptual distinctions among Carver and White's (1994) BAS scales: A reward-reactivity versus trait impulsivity perspective. Personality and Individual Differences, 40(5), 1039–1050. https://doi.org/10.1016/j.paid.2005.10.012

Comments 3: The literature review would benefit from additional studies, as those by Andrea Creech,  on music activities for older adults.

Just three examples:

Creech, A., et al. (2013). The power of music in the lives of older adults. Research Studies in Music Education, 35(1).

Creech, A., et al.  (2013). Active music making: a route to enhanced subjective well-being among older people. Perspectives in Public Health, 133(1).

Laes, T., & Creech, A. (2023). Re-conceptualizing music education in the older adult life course: A qualitative meta-synthesis. International Journal of Population Studies, 9(3).

This could enrich the theoretical framework and strengthen the argument.

Response 3: Thank you for your valuable recommendation regarding the inclusion of additional literature by Andrea Creech and colleagues on music activities for older adults. We have revised and enriched the theoretical background and literature review accordingly, integrating relevant content based on the suggested references. These additions help to strengthen the framework and contextual grounding of our study. [Line 46-55]

Enjoying playing musical instruments is beneficial even in old age. Numerous studies have shown that active music-making enhances older adults’ cognitive, emotional, and social wellbeing (Creech, Hallam, McQueen et al., 2013a; Creech, Hallam, Varvarigou et al., 2013). For example, regular participation in musical activities has been associated with higher levels of subjective wellbeing, self-esteem, and life satisfaction. Moreover, a qualitative meta-synthesis suggested that music education can offer meaningful and empowering experiences in later life stages (Laes & Creech, 2023). In addition, playing musical instruments may help prevent cognitive decline, including dementia, by improving verbal memory and the efficiency of the nervous system (Guo et al., 2021; Arafa et al., 2022).

Creech, A., Hallam, S., McQueen, H., & Varvarigou, M. (2013). The power of music in the lives of older adults. Research Studies in Music Education, 35(1), 87–102. https://doi.org/10.1177/1321103X13478862

Creech, A., Hallam, S., Varvarigou, M., McQueen, H., & Gaunt, H. (2013). Active music making: A route to enhanced subjective well-being among older people. Perspectives in Public Health, 133(1), 36–43. https://doi.org/10.1177/1757913912466950

Laes, T., & Creech, A. (2023). Re-conceptualizing music education in the older adult life course: A qualitative meta-synthesis. International Journal of Population Studies, 9(3), 15–32. https://doi.org/10.36922/ijps.383

Comments 4: The conclusions generally reflect the findings, but a few statements overreach the correlational nature of the study. The distinction between association and causation should be made clearer, and the limitations deserve more consideration when forming the final claims.

Response 4: Thank you for your valuable comment regarding the distinction between correlation and causation in the conclusion section. In response, we have carefully revised the conclusions to avoid any implication of causality and to clearly emphasize the correlational nature of our findings. Specifically, we rephrased statements that may have overreached by incorporating language such as “associated with” and “correlated with,” and included an explicit acknowledgment of the study’s methodological limitations in inferring causality. Additionally, we expanded the conclusion into three paragraphs to clearly distinguish the key findings, offer cautious interpretations, and address implications within the scope of a correlational design. We believe these revisions now reflect a more balanced and rigorous interpretation of our results. [Line 509-523]

We found that extraversion, conscientiousness, agreeableness, the BAS, the BIS, and hardiness were significantly correlated with the enjoyment in playing musical instruments among Chinese adults. Additionally, fun-seeking, agreeableness, openness, and self-directedness were notable predictors of adults’ enjoyment of playing musical instruments. The decision-tree model further identified variables such as enjoyment of music classes in childhood, self-directedness, age, family music experiences, and familial musical atmosphere as being associated with adults’ musical enjoyment. These results indicate patterns of association rather than causation as this study employed a correlational design.

Although traits such as fun-seeking and the BAS/BIS were negatively associated with musical enjoyment, these findings should be interpreted with caution. Individuals scoring high on these traits may find greater satisfaction with less conventional musical activities, such as improvisation, ensemble play, or high-stimulation musical environments. However, these hypotheses require empirical validation. Importantly, the cross-sectional nature of the data limits causal inferences, and reverse or reciprocal relationships cannot be ruled out.

Given these limitations, these findings provide insight that may help music educators and therapists better tailor their approaches. For example, considering individuals’ early music exposure, self-directedness, and motivational tendencies may enhance engagement and outcomes. However, practical applications should be grounded in further longitudinal or experimental research to more clearly disentangle causality from correlation.

Comments 5: The English is generally understandable, but there are recurrent issues with clarity and sometimes with grammar also.

Some examples:

  • In the five paragraphs after the table, number 4 (within 3.2), the phrase "(did not) enjoy music classes in childhood", is repeated eight times, making the text "heavy" in some sense.
  • Another expression that is repeated too much is "this study." While understandable, it does not help the text's fluidity.
  • The paragraph just before 2. "Material and Methods", starting with If we can predict individuals’ ...", is wordy and repetitive, making it confusing.
  • In section 3.2, the sentence ""Openness may positively affect the enjoyment of playing musical instruments in the regression model, although there was no significant correlation between openness and the enjoyment of playing instruments in the correlation analysis", is logically confusing; clarify what you want to say.
  • Same on the page 13 (within the Discussion): "For example, teaching children or adolescents with high fun seeking to play musical instruments or music therapy using musical instruments for individuals with high fun seeking may be ineffective." This is a strange sentence.

I consider that a professional language revision is needed to improve the clarity and impact of the manuscript.

Response 5:

We sincerely appreciate the reviewer’s detailed feedback regarding the language quality and clarity of the manuscript. We acknowledge that, although we previously had the manuscript professionally edited by Editage prior to initial submission, some expressions and sentence structures still lacked clarity and fluency in academic English.

In response to the reviewer’s specific comments:

  • Repetitive phrasing such as "(did not) enjoy music classes in childhood" and "this study" has been revised to enhance stylistic fluidity and avoid redundancy.
  • The paragraph before Section 2 (“Materials and Methods”) has been significantly shortened and reworded to improve clarity and avoid unnecessary repetition.
  • The sentence in Section 3.2 regarding openness has been rewritten to clarify the statistical distinction between bivariate and multivariate analysis.
  • The sentence on page 13 regarding fun-seeking has also been restructured for clarity and logical consistency.
  • Other minor grammar and phrasing issues across the manuscript have been addressed as well.

To further ensure language quality, we submitted the revised manuscript once again to Editage for professional English editing before submitting this revision. A language editing certificate from Editage has been included as supplementary material.

Thank you again for pointing out these critical issues that helped us improve the manuscript’s readability and scholarly tone. [Line 324-347, 136-142, 293-296, 436-437 etc..]

The first criterion used to classify the level of enjoyment in playing musical instruments was whether the participants enjoyed music classes during childhood (Figure 1). Among the 186 participants who responded affirmatively, the average enjoyment score was relatively high at 60.77 (Node 1). Within this group, those with self-directedness scores above 18 reported even higher enjoyment, averaging 65.95 (Node 5). Of these, individuals who also grew up in families that appreciated music showed the highest enjoyment scores, reaching 69.95 (Node 11).

Participants with favorable early music experiences and self-directedness scores between 14 and 18 still showed relatively high enjoyment levels (M = 58.66; Node 4). In this subgroup, those who had played instruments with family members scored higher, averaging 64.55 (Node 10).

In contrast, among the 230 participants who did not report enjoying music classes during childhood, the overall enjoyment score was lower, at 39.17 (Node 2). However, within this group, younger participants (aged 31 years or below) demonstrated a comparatively higher enjoyment score of 55.97 (Node 6).

Among those aged 31–58 years with no early positive music experience, the average score dropped to 37.27 (Node 7). Nevertheless, individuals in this subgroup with agreeableness scores above 16 reported moderately higher enjoyment, at 53.50 (Node 15).

Understanding who is likely to enjoy playing musical instruments can help identify suitable candidates for instrumental training and improve the effectiveness of such programs. Previous research has explored the predictors of participation in music lessons based on parental traits (Corrigall & Schellenberg, 2015). Anticipating musical enjoyment may also inform efforts to enhance wellbeing through music. This insight can help parents and educators avoid pressuring those unlikely to enjoy instrument learning and guide the appropriate use of music-based interventions, such as therapy.

Although openness was not significantly correlated with the enjoyment of playing musical instruments in the bivariate correlation analysis, it emerged as a significant predictor in the stepwise regression model when controlling for other variables.

Therefore, educators and therapists might consider integrating more engaging or novelty-rich elements in early-stage instruction.

Reviewer 2 Report

Comments and Suggestions for Authors

The argument is deterministic and the authors aren't questioning their assumptions rigorously enough. More information is needed on the therapeutic prescribing of musical instrument learning. Music therapy means very different things in the Western world.

On page 3 the authors assert that 'hardy people are more likely to learn musical instruments' ... how about if we ask 'does the discipline of learning an instrument build or strengthen hardiness of character?'

The paper would be strengthened hennaed, and have wider reach, if the concept of music therapy it mentions is described in more detail. Many people are working on music & wellbeing, and community music, and it would help to connect with them.

Author Response

Response to Reviewer 2 Comments

Thank you very much for taking the time to review this manuscript. Please find the detailed responses below and the corresponding revisions/corrections highlighted/in track changes in the re-submitted files. The revised parts were marked in red, and we included the page and line of the revised part. We appreciate your complimentary comments. We have omitted our response to your kind words here.

Point-by-point response to Comments and Suggestions for Authors

Comments 1: The argument is deterministic and the authors aren't questioning their assumptions rigorously enough.

Response 1: Thank you for your valuable comment regarding the determinism in our argument and the need for greater scrutiny of underlying assumptions.

In response, we have carefully revised the conclusion section to adopt a more cautious and balanced tone. Specifically, we have replaced or qualified deterministic language with expressions that more clearly reflect the correlational nature of our findings (e.g., “associated with” instead of “predicted,” “indicate patterns of association,” and “should be interpreted with caution”). We have also explicitly acknowledged the limitations of causal inference due to the cross-sectional design and highlighted the need for further longitudinal or experimental studies to establish causality. These revisions aim to clarify that our interpretations are exploratory rather than confirmatory, and that we recognize the complexity and multi-directionality of psychological and behavioral influences related to musical enjoyment. We hope these changes address your concern and contribute to a more nuanced and rigorous presentation of our findings. [Line 509-523]

We found that extraversion, conscientiousness, agreeableness, the BAS, the BIS, and hardiness were significantly correlated with the enjoyment in playing musical instruments among Chinese adults. Additionally, fun-seeking, agreeableness, openness, and self-directedness were notable predictors of adults’ enjoyment of playing musical instruments. The decision-tree model further identified variables such as enjoyment of music classes in childhood, self-directedness, age, family music experiences, and familial musical atmosphere as being associated with adults’ musical enjoyment. These results indicate patterns of association rather than causation as this study employed a correlational design.

Although traits such as fun-seeking and the BAS/BIS were negatively associated with musical enjoyment, these findings should be interpreted with caution. Individuals scoring high on these traits may find greater satisfaction with less conventional musical activities, such as improvisation, ensemble play, or high-stimulation musical environments. However, these hypotheses require empirical validation. Importantly, the cross-sectional nature of the data limits causal inferences, and reverse or reciprocal relationships cannot be ruled out.

Given these limitations, these findings provide insight that may help music educators and therapists better tailor their approaches. For example, considering individuals’ early music exposure, self-directedness, and motivational tendencies may enhance engagement and outcomes. However, practical applications should be grounded in further longitudinal or experimental research to more clearly disentangle causality from correlation.

Comments 2: The paper would be strengthened hennaed, and have wider reach, if the concept of music therapy it mentions is described in more detail. Many people are working on music & wellbeing, and community music, and it would help to connect with them.

Response 2: Thank you for your thoughtful comment. We have now revised the manuscript to include a clearer and more nuanced explanation of the concept of music therapy, acknowledging the diversity of its interpretations and applications. Specifically, we added a paragraph distinguishing between clinical music therapy and community music therapy, and cited key works that provide a comprehensive framework of music's role in health and wellbeing, including Bruscia (2014), Ansdell & DeNora (2016), and MacDonald (2013). This revision clarifies the therapeutic scope within which instrument-playing is discussed in the manuscript and strengthens the conceptual grounding of our claims. [Line 481-490]

We broadly refer to “music therapy” as the psychological application of instrument-playing. However, music therapy involves diverse approaches. Clinical music therapy involves individualized interventions by licensed professionals, targeting goals such as anxiety reduction or communication enhancement (Bruscia, 2014). In contrast, community music therapy emphasizes inclusive participatory music making that fosters social connection and empowerment regardless of diagnosis (Ansdell & DeNora, 2016). MacDonald (2013) proposed a broader framework in which music supports emotional regulation, cognitive engagement, and social wellbeing. These perspectives highlight the multifaceted therapeutic roles of music and suggest the need for greater clarity when applying or evaluating instrument-based interventions.

Comments 3: On page 3 the authors assert that 'hardy people are more likely to learn musical instruments' ... how about if we ask 'does the discipline of learning an instrument build or strengthen hardiness of character?

Response 3: Thank you for this thoughtful suggestion. We agree that the relationship between hardiness and learning to play a musical instrument may be bidirectional. In response, we have added a sentence in the Discussion section to consider the possibility that the sustained practice and discipline involved in musical training may in fact strengthen psychological hardiness. [Line 459-467]

While we assumed that individuals with higher levels of hardiness tend to enjoy playing musical instruments, it is also worth considering the reverse possibility: whether the discipline and perseverance required to learn an instrument might, in turn, foster psychological hardiness. Long-term musical training involves coping with challenges, maintaining self-directed practice, and persisting through frustration and qualities that overlap with the dimensions of hardiness such as commitment, control, and challenge (Kobasa, 1979b). Future longitudinal research should explore whether structured musical training contributes to the development of hardy traits, particularly among younger learners and clinical populations.
